# Survival of Patients with Epidermal Growth Factor Receptor-Mutated Metastatic Non-Small Cell Lung Cancer Treated beyond the Second Line in the Tyrosine Kinase Inhibitor Era

**DOI:** 10.3390/cancers13153887

**Published:** 2021-08-02

**Authors:** Valéry Refeno, Michele Lamuraglia, Safae Terrisse, Clément Bonnet, Clément Dumont, Ludovic Doucet, Damien Pouessel, Stephane Culine

**Affiliations:** 1Oncology Department, CHU Amiens, Université de Picardie Jules Vernes, 80000 Amiens, France; refeno.valery@chu-amiens.fr; 2Oncology Department, Professeur Zafisaona Gabriel Hospital, Université de Mahajanga, Mahajanga 401, Madagascar; 3Faculté de Médecine, Université d’Antananarivo, Antananarivo 101, Madagascar; 4Sorbonne Universités, UMPC Univ. Paris 06, UMR 7371, UMR S 1146, Laboratoire d’Imagerie Biomédicale, 75005 Paris, France; 5AP-HP, Hôpital Saint Louis, Oncology Unit, 1 Avenue Claude Vellefaux, 75010 Paris, France; safae.terrisse@aphp.fr (S.T.); clement.bonnet@aphp.fr (C.B.); clement.dumont@aphp.fr (C.D.); stephane.culine@aphp.fr (S.C.); 6Department of Medical Oncology, Institut de Cancérologie de l’Ouest, 44800 Saint-Herblain, France; Ludovic.Doucet@ico.unicancer.fr; 7Department of Medical Oncology, Institut Claudius Regaud, IUCT-O, 31300 Toulouse, France; Pouessel.Damien@iuct-oncopole.fr

**Keywords:** EGFR-genes, drug therapy, lung cancer, metastasis, survival analysis, TKI

## Abstract

**Simple Summary:**

The identification of activating mutations in specific genes in non-small cell lung cancer (NSCLC) has led to the development of targeted therapies, which are currently part of the algorithm for their management. The recommendations agree on first and second-line metastatic treatments in Epidermal Growth Factor Receptor (EGFR) mutations patients. The aim of our retrospective, longitudinal and analytic study was to analyze the survival of EGFR-mutated patients treated beyond the second line of treatment. We confirmed in a population of 31 patients which received at least three lines of treatment that the Progression Free Survival (PFS) was best if we used chemotherapy in second-line and tyrosine kinase inhibitors (TKI) in third-line. We found no difference in Overall Survival (OS) according to the pattern of treatments. In practice, in the TKI era, chemotherapy can still be used in second-line or third-line of treatment.

**Abstract:**

Background: The identification of activating mutations in specific genes led to the development of targeted therapies for NSCLC. TKI directed against EGFR-mutations were the first to prove their major efficacy. Medical associations recommend their use as first and second-line metastatic treatments in EGFR-mutated patients. Our objective was to analyze the survival of EGFR-mutated patients treated beyond the second line of treatment. Methods: We performed a longitudinal, retrospective and analytical study at APHP (Assistance Publique Hopitaux de Paris) Saint Louis, Paris, France, from 1 January 2010 to 31 December 2020 (11 years), on EGFR-mutated patients with metastatic NSCLC which received TKI or chemotherapy (CT) in third-line. Results: Out of about 107 EGFR-mutated patients, 31 patients who benefited from TKI or CT in the third line of treatment were retained for this study. The mean age was 60.03 ± 11.93 years and the sex ratio male/female was 0.24. Mutations of exon 19, 21 and 20 were found in 21 (67.7%), 7 (22.6%) and 7 (22.6%) patients, respectively. Third-line treatment was CT for 16 patients (51.6%) and TKI for the 15 remaining patients (48.4%). Osimertinib was the most used TKI in third-line (*n* = 10/15; 66.67%). The median duration of third-line treatment was 5.37 months (range 0.53–37.6) and the median follow-up duration was 40.83 months (range 11.33–88.57). There was a significant difference in PFS between patients treated with TKI and CT in third-line (*p* = 0.028). For patients treated with CT in second-line, there was a significant difference of PFS (*p* < 0.001) and OS (*p* = 0.014) in favor of the use of TKI in third-line. Conclusions: For patients receiving CT in second-line, TKI appears to be a better alternative in third-line compared to CT. Osimertinib may be used in third line treatment if not used before.

## 1. Introduction

In 2020, lung cancer was still the leading cancer worldwide in terms of mortality. NSCLC accounts for nearly 85% of these cancers [1]. In the era of conventional cytotoxic drugs, the prognosis for this cancer was grim [2]. The identification of activating mutations in specific genes has paved the way for the development of targeted therapies, which are currently part of the algorithm for the management of these cancers. TKI directed against EGFR-mutations were the first to prove their major efficacy in the population with these mutations. The recommendations agree on first and second-line metastatic treatments in EGFR-mutated patients [3,4]. Those recommendations are based on results of most relevant study. In 2018, the FLAURA trial demonstrated that Osimertinib in the first line improved PFS and 18-month survival rate compared to standard EGFR-TKIs [5]. In the same year, Imai et al. demonstrated the efficacy and safety of CT as a second line after first-line EGFR-TKI for elder patients with EGFR-mutated NSCLC [6]. To our knowledge, few studies have investigated systemic treatments beyond the second line in this population [3,4]. Thus, the objective of this study was to analyze the survival of EGFR-mutated patients treated beyond the second line of treatment.

## 2. Methods

We performed a longitudinal, retrospective and analytical study at the APHP Saint Louis, Paris, France, from 1 January 2010 to 31 December 2020 (11 years), on EGFR-mutated patients with NSCLC. We included patients with an EGFR mutation. Then, we excluded the unusable files, the patients who never benefited from metastatic systemic treatments, those who received anticancer CT in the first line of treatment, those who did not receive third-line treatment and those who have had third-line immunotherapy. We selected metastatic NSCLCs who received TKI as the first line of treatment. The patients were classified into 4 groups according to the sequence of treatment lines. In sequence 1, the patients successively received TKI in first-line, second-line and third-line treatment (TKI-TKI-TKI). In sequence 2, patients successively received TKI, TKI, and CT (TKI-TKI-CT). In sequence 3, patients successively received TKI, CT, and TKI (TKI-CT-TKI). In sequence 4, patients successively received TKI, CT and CT (TKI-CT-CT).

This study was conducted in accordance with the Declaration of Helsinki, the International Conference on Harmonization harmonized Tripartite Guidelines for good clinical practice, and site-specific institutional review boards approved the study protocol and amendments. The protocol was approved by our local Ethics Committee and our Institutional Review Board.

To estimate the survival of patients, we used the Kaplan–Meier estimator. We used the Log-rank test to compare survival of subgroups according to factors by *p*-value analysis. We considered that there was significant difference between survival of compared subgroups if the *p*-value was inferior to 0.05. The primary endpoints were progression-free survival (or PFS, which is defined as the time between onset of third-line and confirmed disease progression or death) and overall survival (or OS, which is defined as the time between the date of diagnosis of the disease and the date of death or the latest news).

Data were collected on Microsoft Access version 2016^®^, then processed on Microsoft Excel version 2007^®^ and IBM Statistical Package for Social Studies version 20.

## 3. Results

### 3.1. Description of the Study Population

Among the 107 EGFR-mutated patients included, we excluded 76 patients. In the end, we retained 31 patients who benefited from TKI or CT in the third line of treatment. Patient selection is shown in Figure 1.

The mean age was 60.03 ± 11.93 years and the gender ratio was 0.24. Twenty-three patients (74.2%) were non-smokers and the same number had co-morbidities. On initial biomolecular analysis, the deletion of exon 19 was found in 21 patients (67.7%), the L858R mutation of exon 21 in 7 patients (22.6%), the T790M mutation of exon 20 in 7 patients (22.6%) and the exon 18 mutation in 5 patients (16.1%). Note that 10 out of 31 patients (32.26%) presented with at least two associated mutations. The median number of lines received was 3 (range 3–7). Third-line treatment was CT for 16 patients (51.6%) and TKI for the remaining 15 (48.4%). Out of those 15 patients, ten (66.67%) received Osimertinib as third treatment. According to the treatment, the TKI-TKI-CT pattern (*n* = 10; 32.3%) and the TKI-CT-TKI pattern (*n* = 9; 12.7%) were the most represented. In the latter regimen, the TKIs used were Osimertinib (*n* = 7/9; 77.8%) and Erlotinib (*n* = 2/9; 22.2%). The TKI-TKI-TKI and TKI-CT-CT pattern were equally represented (*n* = 6; 19.35%). Note that no patient received Osimertinib as first-line treatment. Among 10 patients who received TKI in the second-line, the CT used in the third line was a combination of a platinum salt with pemetrexed ± bevacizumab for 9 patients (90%) and paclitaxel for the last patient. For the 6 patients who received second-line CT, third-line CT was either docetaxel (*n* = 4; 66.8%) or the combination of paclitaxel with carboplatin (*n* = 2; 33.3%). The median duration of third-line was 5.4 months (range 0.53–37.6). The median duration of follow-up was 40.8 months (range 11.33–88.6). At the end of the study, 23 patients (74.2%) had died, 6 (19.3%) were alive and 2 (6.4%) were lost to follow-up. The epidemioclinical characteristics of patients treated with TKI or CT as third-line were comparable except for age and smoking status. The characteristics of the patients according to the treatment chosen in third-line are reported in Table 1.

### 3.2. Analysis of PFS in Third-Line

The median PFS was estimated at 6 months. There was a significant difference in PFS between patients treated with TKI and CT in third-line (*p* = 0.028). The median PFS was 9 months for people treated with TKI and 3.3 months for those treated with CT (Figure 2a). There was also a significant difference in PFS depending on the treatment sequence (*p* = 0.001). The median PFS was 8.9 months, 6 months, 5.4 months and 0.7 months, respectively, for the TKI-CT-TKI, TKI-TKI-CT, TKI-TKI-TKI and TKI-CT-CT pattern (Figure 2b). For patients treated with CT in second-line, there was a significant difference in PFS in favor of the use of TKI in third-line (*p* < 0.001) (Figure 2c). There was no difference in PFS regardless of the third line chosen in patients who received TKI in second-line (*p* = 0.588) (Figure 2d). For patients who received TKI in third-line, there was no difference in PFS depending on the treatment received in second-line (*p* = 0.443) (Figure 2e). In patients treated with CT in third-line, there was a significant difference in PFS in favor of the use of TKI in second-line (*p* = 0.017) (Figure 2f).

There was no difference in PFS in third-line depending on the presence or absence of mutations in exon 21 (*p* = 0.495), exon 19 (*p* = 0.862) or exon 20 (*p* = 0.903).

### 3.3. Analysis of Overall Survival

The median OS was estimated at 48 months. Disregarding the second line, there was no significant difference in OS whether CT or TKI was used in third-line (*p* = 0.695) (Figure 3a). There was no significant difference in OS depending on the sequence pattern (*p* = 0.125) (Figure 3b). For the patients who had CT in second-line, we found a significant difference in the OS in favor of TKI in third-line (*p* = 0.014). Median OS was 59.3 months for patients treated by the TKI-CT-TKI regimen vs. 34 months for patients who were treated by the TKI-CT-CT regimen (Figure 3c). For patients who had TKI in second-line (TKI-TKI-TKI and TKI-TKI-CT pattern), there was no significant difference in OS regardless of the type of treatment used in third-line (*p* = 0.525) (Figure 3d). For patients who received TKI in third-line, there was no difference in OS depending on the treatment received in second-line (*p* = 0.935) (Figure 3e). In patients treated with CT in third-line, there was no difference in OS depending on the treatment received in second-line (*p* = 0.110) (Figure 3f).

Of the 31 patients, 7 patients presented with an exon 21 mutation. Regardless of the sequence pattern, we found a difference in OS depending on the presence or absence of the exon 21 mutation (*p* = 0.022). The median OS was 59.3 months for patients without an exon 21 mutation versus 47.2 months for patients with it (Figure 4). In patients with a mutation of exon 21 (*n* = 7), there was no difference in OS depending on the choice of third-line (*p* = 0.930). In the subgroup with no exon 21 mutation, there was no significant difference in survival depending on the choice of third-line (*p* = 0.715). There was no significant difference in OS depending on the presence or absence of exon 19 deletion (*p* = 0.756). Likewise, there was no significant difference in OS depending on the presence or absence of the exon 20 mutation (*p* = 0.406).

## 4. Discussion

In our study, the treatment sequence leading to the longest PFS in third-line was the TKI-CT-TKI pattern (8.9 months). For patients who had CT in second-line, we found a significant benefit in terms of PFS (*p* < 0.001) and OS (*p* = 0.014) from the use of TKI in third-line compared to CT. The relevance of this approach was investigated by Song et al. in a study that investigated the efficacy and safety of introducing Gefitinib as third-line in patients initially treated with Gefitinib as first-line and with CT as second-line. Median PFS in the third line in Song et al.’s study was 4.4 months [7]. The renewed susceptibility to TKIs after CT treatment seems to be explained by the co-existence of two cellular contingents in the tumor tissue: one sensitive and one resistant to TKI. In response to the administration of TKIs in first-line, the susceptible fraction is eradicated selecting TKI-resistant clones, which will proliferate and lead to clinical progression of the disease. CT in second-line will act on TKI-resistant fractions, sparing TKI-sensitive cells, which will be accessible to a new TKI treatment [7,8]. In our study, the significant response obtained by the use of TKIs in third-line after CT could also be explained by the fact that Osimertinib was only used after CT and was the main TKI used in third-line (77.78%). The main mechanism of resistance of EGFR-mutated NSCLCs to first and second generation TKIs has been shown to be the acquisition of T790M mutation at exon 20 [9]. Osimertinib, a third generation anti-EGFR TKI was initially developed specifically for patients who presented the T790M mutation and who progressed after the first and second generation anti-EGFR TKIs (AURA trial) [10]. Our data agree with data from Auliac et al., who had shown that Osimertinib has similar efficacy in clinical studies and in real life data in terms of PFS and OS whether introduced as second-line or third-line treatment in patients with a T790M mutation progressing after first or second generation anti-EGFR and CT [11]. Their effectiveness in our study is all the more justified by the fact that three patients in this subgroup already had a T790M mutation at diagnosis. In our study, Osimertinib was not used in third-line in the two remaining patients nor in the first line for patients who had T790M-mutation at diagnosis because the drug was not yet available at this time. According to ASCO/CCO 2020 recommendations, Osimertinib should be used as the first line in EGFR-mutated patients diagnosed with T790M, L858R mutations or exon 19 deletion. Other anti-EGFR TKIs may be used if Osimertinib is not available. In the absence of these mutations, Afatinib appears to be the best first-line alternative to Osimertinib. In the event of progression under TKI without the T790M mutation or in the event of progression under Osimertinib, CT may be offered as a second line. To date, there are no recommendations on what to do with progression after TKI and CT [3]. In the third line, after first or second generation TKI and CT, resuming TKI (third generation) seems to be the best third-line option compared to CT. Our conclusions are limited when it comes to their extrapolation in a population treated first-line with Osimertinib and then with CT. The development of resistance to Osimertinib includes both EGFR-dependent and EGFR-independent mechanisms. Multiple strategies are being studied, including the combination of Osimertinib with other TKIs, CT or immunotherapy. Some authors also raise the hypothesis of alternating sequential treatment with different TKIs, each of them acting on clones resistant to other TKIs. Other authors suggest the path of sequential dosing of treatments to try to establish a balance between sensitive and resistant clones [12,13,14,15]. Actually, it was observed the ability of metformin to revert resistance to Gefitinib in NSCLC. Moreover, in the phase I-II trial METformin in Advanced Lung cancer (METAL), metformin combined with erlotinib in second-line treatment of patients with stage IV NSCLC showed a good safety [16].

In our study, for patients treated with TKI in the second line, we did not find a significant difference between TKI and CT in third-line in terms of PFS (*p* = 0.588) and OS (*p* = 0.525). CT remains an effective treatment in EGFR-mutated NSCLCs and should be used optimally as long as the patient’s condition allows due to their complementary effect to that of TKIs. This was found in the study by Han et al. in China, in which PFS was similar in patients who received TKI in first-line then CT in second-line and in those who received CT in the first line then TKI in the second line (*p* = 0.886) despite the significant difference in PFS in the first line, which is in favor of TKI (*p* < 0.001) [17]. The study by Eriguchi et al. in Japan showed that median OS was better in patients who received CT and TKI compared to those who received only TKI [18]. Therefore, the lack of difference in terms of OS in our study between the TKI-TKI-TKI and TKI-TKI-CT pattern could be explained by the following lines of treatment which included CT. On the other hand, the absence of difference in PFS of these two patterns could be the result of the small size of our sample and would need to be confirmed by other studies before being able to consider the pattern with three successive lines of TKI. Numerous studies (LUX-Lung, GIO TAG study, Tamiya et al.) have studied the successive sequences of anti-EGFR TKI and agree that Afatinib followed by Osimertinib regimen seems to be the best choice for the first two lines of treatment [19,20,21]. We did not find a study that evaluated the effectiveness of using three successive anti-EGFR TKI lines in EGFR-mutated NSCLCs. The TKI-TKI-TKI pattern in our study likely resulted from the gradual onset of different generations of anti-EGFR TKIs over the study period. Pending new data, we suggest applying the ASCO/OH 2020 recommendations that CT is the most relevant treatment for progression after two lines of TKI [3].

In our study, there was no significant difference in PFS in the third line depending on the type of EGFR mutations identified. We found significantly less OS in the presence of the exon 21 mutation. The presence or absence of exon 19 or exon 20 mutations did not give a significant difference in terms of OS. Data from studies are not unanimous on the association between survival and the type of mutation present. In the study by Sutiman et al. in China, there was no significant difference in OS depending on the type of mutation (*p* = 0.054) [22]. In the study by Jiang et al. done in China, in univariate analysis, OS was better in patients with the exon 19 mutation compared to those with the exon 21 mutation (30.2 vs. 25.6 months; *p* = 0.030) [23]. In the study by Lin et al. in the United States, OS was better with a mutation of exon 19 compared to that of exon 18 or 21 (*p* = 0.001) [24]. Studies remain to be done to determine the types of mutations that influence the survival of patients treated beyond the second line of treatment.

The retrospective monocentric design of the study represents a limitation of results analysis. It was not possible to trace the reasons for each patient’s therapeutic algorithm. The small cohort of 31 patients and with even smaller subgroups reduces the statistical strength of the study. Furthermore, the clinical application of the results is limited because in practice the majority of newly diagnosed patients are directly put on first-line Osimertinib.

## 5. Conclusions

For EGFR-mutated patients who received TKI in the first line and CT in the second line, TKI appears to be a better alternative in the third line compared to CT. Osimertinib may be used in third-line treatment if not used before. Our data are not applicable to patients who received Osimertinib in the first line. Nevertheless, the studies in progress are going in the direction of privileging the TKI for the advanced lines. Our study is distinguished by the fact that for patients treated with TKI in the second line, we did not find a significant difference between TKI and CT in the third line in terms of PFS. Due to the lack of studies on the efficacy of three successive lines of anti-EGFR TKI, we suggest using CT after two successive lines of TKI as suggested by international recommendations. Studies remain to be done to determine the types of mutations that influence the survival of patients treated beyond the second line of treatment.

## Figures and Tables

**Figure 1 cancers-13-03887-f001:**
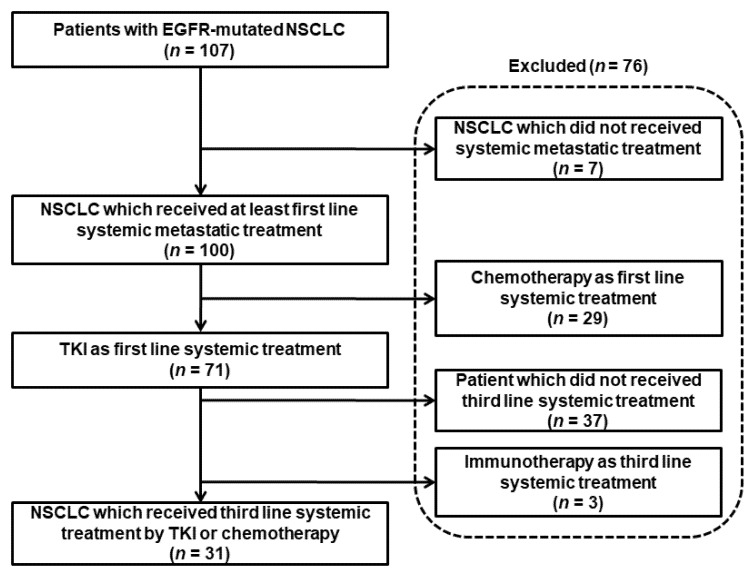
Selection of patients included in the study.

**Figure 2 cancers-13-03887-f002:**
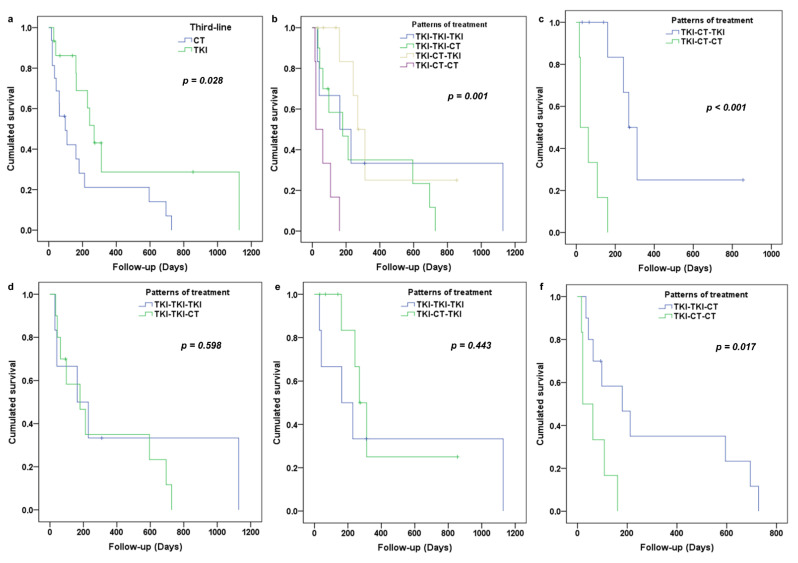
PFS according to the treatment received in third-line (**a**) and according to the pattern of treatment (**b**). PFS according to the choice of third-line for patient which received CT in second-line (**c**) and received TKI in second-line (**d**). (**e**,**f**) represent respective PFS according to choice of second-line for patient which received TKI in third-line and CT in third-line.

**Figure 3 cancers-13-03887-f003:**
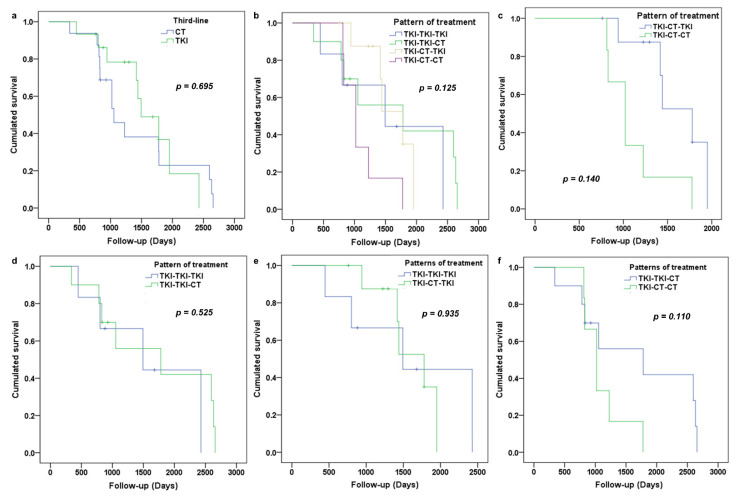
OS according to the treatment received in third-line (**a**), according to the pattern of treatment (**b**), according to the choice of third-line for patient which received CT in second-line (**c**) and according to the choice of third-line for patient who received TKI in second-line (**d**). (**e**,**f**) represent respective OS according to choice of second-line for patient which received TKI in third-line and CT in third-line.

**Figure 4 cancers-13-03887-f004:**
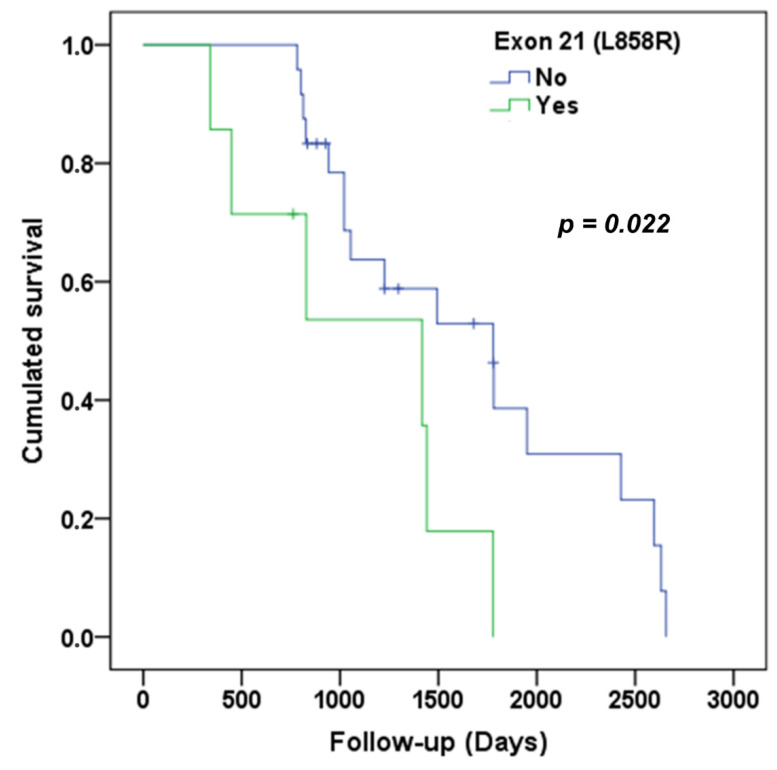
OS according to the presence or absence of the exon 21 mutation.

**Table 1 cancers-13-03887-t001:** Patients’ characteristic according to the third line of treatment.

	Patients Treated by Chemotherapy in Third-Line (*n* = 16)	Patients Treated by TKI in Third-Line (*n* = 15)
Meanage (year)	55.5 ± 11.89	64.93 ± 10.20
Sex-ratio (male/female)	0.23	0.25
Non-smokers [*n* (%)]	9 (56.25)	14 (93.33)
Comorbidities [*n* (%))	12 (75.00)	11 (73.33)
Frequency of EGFR-mutations		
Exon 21(L858R) [*n* (%)]	3 (18.75)	4 (26.67)
Exon 20 (T790M) [*n* (%))	4 (25.00)	3 (20.00)
Exon 19 (délétion) [*n* (%)]	11 (68.75)	10 (66.67)
Exon 18 [*n* (%)]	4 (25.00)	1 (6.67)
Median number of received line of treatment	3.50 (range 3–7)	3.00 (range 3–7)
Median duration of third-line (months)	3.18 (range 0.53–24.27)	7.63 (range 1–37.60)
Median duration of follow-up (months)	34.03 (range 0.53–24.26)	47.23 (range 1–37.6)
Breakdown by state at the latest news		
Deceased [*n* (%)]	14 (87.50)	9 (60.00)
Alive [*n* (%)]	1 (6.25)	5 (33.33)
Lost to follow-up [*n* (%)]	1 (6.25)	1 (6.67)

## Data Availability

The data presented in this study are available upon request from the corresponding author. The data are not publicly available due to regulations of the institution.

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
