# Peer review of "Survival of Patients with Epidermal Growth Factor Receptor-Mutated Metastatic Non-Small Cell Lung Cancer Treated beyond the Second Line in the Tyrosine Kinase Inhibitor Era"

_cancers, 2021, doi:10.3390/cancers13153887_

Round 1

Reviewer 1 Report

In this retrospective study, the authors analyzed treatment outcomes of EGFR mutated lung cancer patients who received 1st, 2nd, and 3rd line therapies (n = 31). No patient received front-line osimertinib. There are four patterns of treatment; TKI-CT-TKI, TKI-CT-CT, TKI-TKI-CT, and TKI-TKI-TKI. The authors conclude that "(Abstract) For patient receiving CT in second-line, TKI appears to be a better alternative in third-line compared to CT for patients who did not receive osimertinib in first-line." The reviewer thinks this conclusion is misleading, because osimertinib may be used in the third line for many of these patients (probably with T790M) as a TKI. 

Similar to the above comment, the choice/efficacies of third line therapy is affected by several factors including,

(1) acquired resistance mechanisms to TKI (especially for the presence/absence of T790M).

(2) the reasons of front-line TKI discontinuation (if front-line TKI is stopped due to adverse effect but not disease progression, high efficacy of 3rd line TKI can be anticipated).

(3) PS and other factors.

Therefore, the reviewer considers that it is difficult to obtain a meaningful results for this topic by a retrospective analysis of a small cohort.

Author Response

ANSWERS R1

1.The authors conclude that "(Abstract) For patient receiving CT in second-line, TKI appears to be a better alternative in third-line compared to CT for patients who did not receive osimertinib in first-line." The reviewer thinks this conclusion is misleading, because osimertinib may be used in the third line for many of these patients (probably with T790M) as a TKI. Similar to the above comment, the choice/efficacies of third line therapy is affected by several factors including,

(1) acquired resistance mechanisms to TKI (especially for the presence/absence of T790M).

(2) the reasons of front-line TKI discontinuation (if front-line TKI is stopped due to adverse effect but not disease progression, high efficacy of 3rd line TKI can be anticipated).

(3) PS and other factors.

Therefore, the reviewer considers that it is difficult to obtain a meaningful results for this topic by a retrospective analysis of a small cohort.  

1.    Thank you for the suggestions. We changed the conclusion.

·       We precised that Osimertinib may be used in third-line. Changes were made in the abstract (LINES 45-47) and in the conclusion section (LINES 258-260).

·       Regarding the other factors that may affect the efficacy/choice of the third-line, we integrated same comments in discussion section (LINES 242-245).

Reviewer 2 Report

In this study, authors have compared the survival of patients with EGFR-mutated metastatic non-small cell lung carcinoma (NSCLC) that were treated beyond the second-line of treatment. Authors have found that patients that were treated with chemotherapy (CT) in the second-line of treatment may have better survival (progression-free survival (PFS) and overall survival (OS)) if they are treated with tyrosine kinase inhibitors (TKI) in the third-line rather than CT. Authors also determined the differences in survival in these patients based on the most common EGFR mutations. The findings from this study thus have important implications in the therapy of NSCLC patients in deciding the sequence of treatments. Here are some suggestions:

1. Abstract: a) Words are missing in several sentences, making it difficult to  understand what authors want to convey. As an example: “Results: about 107 EGFR-mutated patients 31 retained who benefited from TKI or CT in the third-line of treatment. The mean age was 60.03 +/- 11.93 years and the sex ratio was 0.24. Mutations of exon 19, 21 and 20 were found in 30 (67.7%), 7 (22.6%) and 7 (22.6%), respectively.”

Words in italics are missing: Out of about 107 EGFR-mutated patients 31 retained patients who benefited from TKI or CT in the third-line of treatment were retained for this study. The mean age was 60.03 +/- 11.93 years and the sex ratio was 0.24 (not clear M/F or F/M?). Mutations of exon 19, 21 and 20 were found in 30 (67.7%), 7 (22.6%) and 7 (22.6%), patients respectively.

Please proof-read carefully.

b) Mentioning of osimertinib in the conclusions statement is not meaningful since there is no mention of this anywhere else in the abstract.

2. Introduction: Introduction may benefit if some more relevant information is provided for some earlier published clinical trials/studies and their results. For example, authors mention that “The recommendations agree on first and second-line metastatic treatments in EGFR-mutated patients.” The rationale for the analysis presented in this manuscript will be strengthened if details for some of the studies related to the above sentence are provided in the introduction.

3. Methods: It is very important to include a complete statistical section in the methods. How p-values were calculated and other statistical analyses were performed? This information should also be provided in the figure legends.

4. Results:

a) Please provide data- if there was a difference in PFS with any of the EGFR mutations regardless of the sequence of treatments?

b) Figures 2 and 3: use bigger text size for labeling since these are difficult to read and include the p-values in the figures themselves. Also, for figure 4- include p-values in the figure.

c) For OS shown in Figure 3, it will be easier to compare these with PFS if comparisons are made same as in Figure 2. Change “c” to “d” in figure 3 and also provide data for OS survival curves comparisons for TKI-TKI-TKI vs TKI-CT-TKI and TKI-TKI-CT vs TKI-CT-CT.

In addition, minor concerns and suggestions are:

  1. Figure 1: Please change to “NSCLC patients who did not receive” (change “which” to “who” throughout the manuscript and “did not received” to “did not receive”.
  2. Table 1: In the sex ratio- what do “H” and “F” represent?
  3. Figure 2 legend: “E and F represent respectivement the PFS according….”. Remove “ment the”.
  4. “Acknowledgments: The authors gratefully acknowledge the patient and your family…”. Change to "…patients and their family…”.

Author Response

ANSWERS R2

1. Abstract:

a) Words are missing in several sentences, making it difficult to  understand what authors want to convey. As an example: “Results: about 107 EGFR-mutated patients 31 retained who benefited from TKI or CT in the third-line of treatment. The mean age was 60.03 +/- 11.93 years and the sex ratio was 0.24. Mutations of exon 19, 21 and 20 were found in 30 (67.7%), 7 (22.6%) and 7 (22.6%), respectively.”

Words in italics are missing: Out of about 107 EGFR-mutated patients 31 retained patients who benefited from TKI or CT in the third-line of treatment were retained for this study. The mean age was 60.03 +/- 11.93 years and the sex ratio was 0.24 (not clear M/F or F/M?). Mutations of exon 19, 21 and 20 were found in 30 (67.7%), 7 (22.6%) and 7 (22.6%), patients respectively.

Please proof-read carefully.

b) Mentioning of osimertinib in the conclusions statement is not meaningful since there is no mention of this anywhere else in the abstract.

1.    Thank you for the suggestions.

a.     As requested, we added the missing words in the sentences (LINES 34-41) and we proof-read carefully all manuscript with a native English language speaker.

b.    We added a Osimertinib result in abstract (LINES 39).

2. Introduction:

Introduction may benefit if some more relevant information is provided for some earlier published clinical trials/studies and their results. For example, authors mention that “The recommendations agree on first and second-line metastatic treatments in EGFR-mutated patients.” The rationale for the analysis presented in this manuscript will be strengthened if details for some of the studies related to the above sentence are provided in the introduction.

2.    Thank you for comment, at your request we added it in the introduction (LINES 61-65).

3. Methods:

It is very important to include a complete statistical section in the methods. How p-values were calculated and other statistical analyses were performed? This information should also be provided in the figure legends.

3.    Thanks, we added a complete statistical section. (LINES 87-90)

4. Results:

a) Please provide data- if there was a difference in PFS with any of the EGFR mutations regardless of the sequence of treatments?

b) Figures 2 and 3: use bigger text size for labeling since these are difficult to read and include the p-values in the figures themselves. Also, for figure 4- include p-values in the figure.

c) For OS shown in Figure 3, it will be easier to compare these with PFS if comparisons are made same as in Figure 2. Change “c” to “d” in figure 3 and also provide data for OS survival curves comparisons for TKI-TKI-TKI vs TKI-CT-TKI and TKI-TKI-CT vs TKI-CT-CT.

4.    Thank you for your comment:

a)    We were unable to study PFS according to the EGFR mutations because the sous-groups were heterogeneous and small. We precised this limit of study in discussion section (LINES 243-245).

b)   Thank you, for this important suggestion. We have modified the Figure 2, 3 and 4.

c)    We have modified as your request.

In addition, minor concerns and suggestions are:

1.       Figure 1: Please change to “NSCLC patients who did not receive” (change “which” to “who” throughout the manuscript and “did not received” to “did not receive”.

2.       Table 1: In the sex ratio- what do “H” and “F” represent?

3.       Figure 2 legend: “E and F represent respectivement the PFS according….”. Remove “ment the”.

4.       “Acknowledgments: The authors gratefully acknowledge the patient and your family…”. Change to "…patients and their family…”.

Thank you for your suggestions:

1.       We have modified the figure 1.

2.       We have apported the modification in table I.

3.       We have corrected the legend of the Figure 2.

4.       We have corrected the sentence in the acknowledgements section (LINE 336-37).

Reviewer 3 Report

In this study, in patients with non-small cell lung cancer (NSCLC) treated with TKI in second-line, the authors did not find a significant difference between TKI and chemotherapy in third-line in terms of PFS.

The study is interesting and well written. This reviewer raises some issue and advice that the authors have to addressed.

1- A paragraph on study limitations is missing from the text. In particular, the retrospective design of the study represents an important limitation that prevents us from defining a cause-effect relationship. The authors must address this issue.

2- Actually, it was observed the ability of metformin to revert resistance to gefitinib, a selective epidermal growth factor receptor (EGFR) tyrosine kinase inhibitor, in non-small-cell lung cancer (NSCLC). Moreover, in the phase I-II trial METformin in Advanced Lung cancer (METAL), metformin combined with erlotinib in second-line treatment of patients with stage IV NSCLC showed a good safety (ESMO OPEN, 2017, 2 (2): e000132. doi: 10.1136 / esmoopen-2016-000132). This approach could improve survival and overall outcome. The authors should add this issue and these references in the text.

3- A linguistic revision of the text by a native English speaker is suggested.

Author Response

ANSWERSR3

A paragraph on study limitations is missing from the text. In particular, the retrospective design of the study represents an important limitation that prevents us from defining a cause-effect relationship. The authors must address this issue.

1.    Thank you for the suggestions, we added a paragraph on study limitations (LINES 242-246).

1. Actually, it was observed the ability of metformin to revert resistance to gefitinib, a selective epidermal growth factor receptor (EGFR) tyrosine kinase inhibitor, in non-small-cell lung cancer (NSCLC). Moreover, in the phase I-II trial METformin in Advanced Lung cancer (METAL), metformin combined with erlotinib in second-line treatment of patients with stage IV NSCLC showed a good safety (ESMO OPEN, 2017, 2 (2): e000132. doi: 10.1136 / esmoopen-2016-000132). This approach could improve survival and overall outcome. The authors should add this issue and these references in the text.

2.    Thank you for this relevant reference, we added this issue and the reference in the text (LINES 204-207).

2. A linguistic revision of the text by a native English speaker is suggested.

1.    The manuscript was revised by a native English language speaker.

Round 2

Reviewer 1 Report

The reviewer understands that the authors tried to improve their manuscript, however, it is difficult to obtain a meaningful results for this topic by a retrospective analysis of a small cohort.  

They conclude that "For patient receiving CT in second-line, TKI appears to be a better alternative in third-line compared to CT for patients who did not receive Osimertinib in first-line".  However, the reviewer thinks this would be true only if the patient has developed T790M mutation (as AURA3 and other trials have proved it). The current manuscript says nothing if TKI (osimertinib) is better or not for patients without T790M.

Another conclusion by the authors "After two successive lines of TKI, CT should be used." is not supported by data. The reviewer thinks if a patient receives 1st-line gefitinib, 2nd-line erlotinib, and then develops T790M, the 3rd line therapy should be osimertinib but not CT.

Reviewer 2 Report

The authors have responded to all my comments and suggestions satisfactorily.

Reviewer 3 Report

The authors addresses all issuee raised by this reviewer.